# Long-Term Technical Performance of the Osypka QT-5^®^ Ventricular Pacemaker Lead

**DOI:** 10.3390/jcm10040639

**Published:** 2021-02-08

**Authors:** Georg Semmler, Fabian Barbieri, Karin Thudt, Paul Vock, Deddo Mörtl, Harald Mayr, Christian Georg Wollmann, Agne Adukauskaite, Bernhard Pfeifer, Thomas Senoner, Wolfgang Dichtl

**Affiliations:** 1Department of Internal Medicine III, University Clinic St. Pölten & Karl Landsteiner Private University, 3100 St. Pölten, Austria; georg.semmler@meduniwien.ac.at (G.S.); karin.thudt@stpoelten.lknoe.at (K.T.); paul.vock@stpoelten.lknoe.at (P.V.); deddo.moertl@stpoelten.lknoe.at (D.M.); h.mayr@cardio.at (H.M.); christiangeorg.wollmann@stpoelten.lknoe.at (C.G.W.); 2Department of Internal Medicine III, Division of Gastroenterology and Hepatology, Medical University Vienna, 1090 Vienna, Austria; 3Department of Internal Medicine III, Cardiology and Angiology, Medical University of Innsbruck, 6020 Innsbruck, Austria; fabian.barbieri@i-med.ac.at (F.B.); agne.adukauskaite@tirol-kliniken.at (A.A.); thomas.senoner@i-med.ac.at (T.S.); 4Institute of Medical Informatics, UMIT TIROL, Eduard Wallnöfer Zentrum, 6600 Hall in Tirol, Austria; bernhard.pfeifer@umit.at; 5Landesinstitut für Integrierte Versorgung, LIV, Tirol Kliniken GmbH, 6020 Innsbruck, Austria

**Keywords:** pacemaker malfunction, impedance decline, reoperation, lead replacement

## Abstract

Background: Lead-associated complications and technical issues in patients with cardiac implantable electronic devices are common but underreported in the literature. Methods: All patients undergoing implantation of the Osypka QT-5^®^ ventricular lead at the University Clinic St. Pölten between 1 January 2006 and 31 December 2012 were retrospectively analyzed (*n* = 211). Clinical data including pacemaker follow-up examinations and the need for lead revisions were assessed. Kaplan–Meier analysis to estimate the rate of lead dysfunction during long-term follow-up was conducted. Results: Patients were followed for a median of 5.2 years (interquartile range (IQR) 2.0–8.7). R-wave sensing properties at implantation, compared to last follow-up, remained basically unchanged: 9.9 mV (IQR 6.8–13.4) and 9.6 mV (IQR 5.6–12.0), respectively). Ventricular pacing threshold significantly increased between implantation (0.5 V at 0.4 ms; IQR 0.5–0.8) and the first follow-up visit (1.0 V at 0.4 ms; IQR 0.8–1.3; *p* < 0.001) and this increase persisted throughout to the last check-up (0.9 V at 0.4 ms; IQR 0.8–1.2). Impedance significantly declined from 1142 Ω (IQR 955–1285) at implantation to 814 Ω (IQR 701–949; *p* < 0.001) at the first check-up, followed by a further decrease to 450 Ω (IQR 289–652; *p* < 0.001) at the last check-up. Overall, the Osypka QT-5^®^ ventricular lead was replaced in 36 patients (17.1%). Conclusions: This report shows an unexpected high rate of technical issues of the Osypka QT-5^®^ ventricular lead during long-term follow-up.

## 1. Introduction

Bradycardia is a frequently observed clinical issue, commonly associated with syncope, transient dizziness, fatigue and dyspnea on exertion [1] Although cardiac pacing is highly effective in reducing symptoms, morbidity and mortality [2,3,4], short- and long-term complications are frequent in pacemaker carriers [5]. Both lead-associated complications (i.e., dislocations, insulation defects, lead fractures, sensing or threshold issues) and non-lead-associated complications (i.e., infection, myocardial perforation, pneumothorax, lead connection malfunction, hematoma, procedure-related deaths) are commonly encountered, with lead-associated complications being the most frequent ones [6]. Although leadless pacing systems are increasingly used, their broad applicability is limited to few indications and is restricted because of financial constraints due to their high costs [7,8]. Therefore, there is still a high demand in conventional pacemaker systems requiring transvenous leads. In this retrospective study, the long-term technical performance of the bipolar Osypka QT-5^®^ ventricular lead (Osypka, Rheinfelden-Herten, Germany) is reported.

## 2. Methods

### 2.1. Study Design and Patients

Patients undergoing pacemaker implantation at the University Clinic of St. Pölten between 1 January 2006 and 31 December 2012 were screened for eligibility. Only patients with a newly implanted Osypka QT-5^®^ ventricular lead were included. Patient characteristics were extracted from electronic health records including clinical baseline parameters (age, gender, body-mass index (BMI), and co-morbidities) and the pacing indication. Surgery details (daytime, operation time, X-ray time, combination with percutaneous coronary intervention) as well as the used device model, atrial lead model and intraoperative measurements were documented. Peri- and postoperative complications occurring after the implantation were additionally recorded. Patients were followed until death or revision of their Osypka QT-5^®^ ventricular lead. All reoperations during this period were recorded, and specific indications scored as battery-associated, Osypka QT-5^®^ lead-associated, atrial lead-associated, infection-associated, heart failure-associated or other reasons.

### 2.2. Device Follow-Up Examinations

Routine follow-up visits after initial pacemaker implantation were performed according to a standardized protocol. After the first control of the newly implanted device before hospital discharge, visits were scheduled at four weeks and every 12 months thereafter. This interval could be shortened according to the physician’s discretion if the clinical condition of the patient had worsened, if events had occurred presumably being pacemaker-associated or in the case of uncertainty about technical issues. At each visit, technical measurements were performed, including assessment of battery capacity (impedance), atrial lead sensing properties, atrial lead pacing threshold, atrial lead impedance, ventricular lead sensing, ventricular lead pacing threshold and ventricular lead impedance. Lead sensitivity is expressed in millivolts (mV), pacing amplitude is expressed in volts at a pulse width of 0.4 milliseconds (ms) and lead impedance is expressed in ohm (Ω). Only bipolar impedance measurements were considered to allow adequate comparison. The first pacemaker follow-up examination was defined as the check-up before hospital discharge or at 4 weeks in case of missing data on the check-up before hospital discharge. Lead dysfunction was defined either as an increase in pacing threshold above 2.0 V, an impedance ≤200 Ω or ≥1500 Ω and/or a loss of sensing properties leading to impairment of adequate pacing. The decision for lead revision was taken on an individual basis by the treating physician’s discretion.

### 2.3. Statistical Analysis

Continuous variables were reported as median and interquartile range (IQR), categorical variables were shown as numbers of patients and percentage. The distribution of continuous variables was assessed by using the Kolmogorov–Smirnov test and inspection of histograms. The comparisons of continuous variables were performed by either using the paired *t*-test or the Wilcoxon test, according to their distribution. For the assessment of lead dysfunction rate, Kaplan–Meier estimates were calculated. Statistical analysis was conducted using IBM SPSS, version 24 (IBM Corporation, Armonk, NY, USA), graphics were designed using GraphPad PRISM, version 5 (GraphPad Software, Inc., La Jolla, CA, USA). *p*-Values ≤ 0.05 were considered statistically significant.

### 2.4. Ethics

This study was approved by the ethics committee of the University Clinic St. Pölten. Being a retrospective analysis, the requirement of a written informed consent was waived by the ethics committee.

## 3. Results

### 3.1. Patient Characteristics

Overall, 211 patients with a newly implanted Osypka QT-5^®^ ventricular lead were investigated, baseline characteristics are shown in Table 1. The majority of patients were male (*n* = 109, 51.7%), mean age was 77.2 ± 9.9 years, median BMI was 27 (IQR: 24.1–29.4) kg/m^2^. Co-morbidities included arterial hypertension in 162 patients (76.8%), coronary artery disease in 83 patients (39.3%), diabetes mellitus in 42 patients (19.9%), chronic kidney disease in 42 patients (25.1%) and history of stroke in 19 patients (9.0%).

The Osypka QT-5^®^ ventricular lead was used in 16 patients (7.6%) who had already been implanted with a pacemaker before and required a reoperation, whereas in 195 patients, the lead was used undergoing de novo pacemaker implantation. Pacing indications were different forms of atrioventricular block in 79 patients (37.4%), SSS in 47 patients (22.2%), combined atrioventricular block and SSS in 7 patients (3.3%), AF with bradycardia in 55 patients (26.1%) and bradycardia-tachycardia syndrome in 23 patients (10.9%). A dual-chamber pacemaker (DDDR) was used in 138 patients (65.4%) and a single-chamber pacemaker (VVIR) was used in 73 patients (34.6%; Table 2). Several devices were used from different vendors including Sensia^®^ SESR01 (Medtronic, Dublin, Ireland), Adapta^®^ ADDR01 (Medtronic, Dublin, Ireland) and Nexus^®^ I Ultra DR 1490 (Boston Scientific, Marlborough, MA, USA). The 4592 CapSure SP Novus^®^ (Medtronic, Dublin, Ireland) was by far the most frequently implanted atrial lead (Table 2) in case of implantation of a dual-chamber pacemaker. The majority of procedures were performed during the daytime (*n* = 94 (44.5%) between 6 AM and 12 noon, *n* = 77 (36.5%) between 12 noon and 6 PM), while a minority of implanting procedures were done outside of routine working hours. The median operation-time was 29 (21–46) min and the median X-ray time was 2.8 (1.8–4.3) min with 10 pacemaker implantations (4.7%) being combined with a percutaneous coronary intervention. Venous access was from the left side in 197 patients (92.9%), whereas only 15 patients were implanted from the right side.

### 3.2. Perioperative Complications

Perioperative complications within 28 days after pacemaker implantation occurred in 23 patients (10.9%), with the development of a postoperative hematoma being the most prevalent occurring in 9 patients (4.3%). Seven patients (3.3%) suffered from a subclinical or oligosymptomatic pneumothorax; all of them could be treated conservatively. Early reoperation and revision of the Osypka QT-5^®^ ventricular lead was needed in 7 patients (3.3%): in 4 patients (1.9%) because of a dislocation, in 2 patients because of an infection (0.9%) and one patient because of a loose set screw (0.5%). Post-operative mortality within 28 days (*n* = 6, 2.8%) was not directly related to the pacemaker implantation but due to a recent myocardial infarction (*n* = 3, 1.4%), stroke (*n* = 1, 0.5%), acute heart failure (*n* = 1, 0.5%) and another severe neurological disease (*n* = 1, 0.5%).

### 3.3. Long-Term Outcomes

Overall, patients were followed for a median of 5.2 years (IQR 2.0–8.7). During follow-up, 106 patients (50.2%) died resulting in a median survival rate of 7.8 years. Five-years survival rate was 62.8%. No death was documented to be directly linked to any pacemaker complication or dysfunction.

### 3.4. Long-Term Technical Performance of the Osypka QT-5^®^ Ventricular Lead

The majority of patients (*n* = 185, 87.7%) received regular pacemaker follow-up examinations at our center, while 26 patients (12.3%) received pacemaker follow-up examinations elsewhere. Overall, patients received a median of four follow-up examinations (IQR 1–11) with a maximum of 24 control visits. The median R-wave sensing was 9.9 mV (6.8–13.4) at implantation and 9.6 mV (5.6–12.0) at last follow-up examination (Figure 1). The ventricular pacing threshold significantly increased between implantation (0.5 V at 0.4 ms; IQR 0.5–0.8) and the first follow-up visit (1.0 V at 0.4 ms; IQR 0.8–1.3; *p* < 0.001) and this increase persisted throughout to the last check-up (0.9 V at 0.4 ms; IQR 0.8–1.2). Impedance significantly declined from 1142 Ω (IQR 955–1285) to 814 Ω (IQR 701–949; *p* < 0.001) at the first check-up and 450 Ω (IQR 289–652; *p* < 0.001) at the last check-up. This resulted in a median difference of minus 252 Ω (IQR −453 to −158) between pacemaker implantation and first check-up, and of minus 286 Ω (IQR −472 to −120) between the first and the last check-up. When only considering patients with less than 5 years of follow-up, the median difference between the first and last pacemaker check-up was minus 119 Ω (IQR −295 to 34) at a median last impedance of 659 Ω (IQR 527 to 869). Regarding patients with at least 5 years of follow-up, it was minus 384 Ω (IQR −546 to −250) at a median last impedance of 390 Ω (IQR 267 to 522; *p* < 0.001).

### 3.5. Long-Term Pacemaker Reoperation and Osypka QT-5^®^ Ventricular Lead Explantation Rates

During a median follow-up period of 4.0 years (IQR 0.1–8.0), 78 patients (37.0%) required reoperation (Table 3), with the major indication being battery replacement due to depletion in 36 patients (17.1%). Other indications included upgrading to a biventricular device in 10 patients (4.7%), complete retrieval of the pacemaker system because of infection in 2 patients (0.9%) or repositioning of atrial and/or ventricular leads due to dislocation in 7 (3.3%) patients. Battery replacement with precautionary revising the ventricular lead due to low impedance was performed in 5 (2.4%) patients.

Overall, the Osypka QT-5^®^ ventricular lead was revised in 36 patients (17.1%). Lead dysfunction requiring revision was observed in 11 patients (5.2%) during long-term follow up. The Kaplan–Meier curve describing the estimates over time is displayed in Figure 2. Specific patient data are further displayed in Table 4.

## 4. Discussion

To the best of our knowledge, this is the first study to analyze the long-term performance of the Osypka QT-5^®^ ventricular lead in a large cohort of patients with predominately on-site, strict check-up examinations. The lead itself is a passive-fixation straight transvenous bipolar pacing lead with an insulation made of polyurethane, which is at increased risk of environmental stress cracking and oxidation, potentially leading to early insulation failure [9,10,11]. Despite the gradual decrease in lead impedance, which is usually regarded as an indicator for insulation problems and incipient lead fracture often leading to concomitant severe pacing problems, patients with Osypka QT-5^®^ leads showing low impedance and being treated conservatively over several years did not yield any serious clinical complications. The slight but significant increase in pacing threshold did not affect output programming and battery longevity, and was detected only very late in our analysis. Indeed, many unnecessary lead revisions could be avoided by our principal “wait and see” strategy.

Although there was a significant rate of lead revisions (17.1%), only 5.2% were due to prevailing lead dysfunction. Fortunately, no deaths were directly associated with any pacemaker complication and/or dysfunction. In comparison to the literature, the rate of lead dysfunction seems to be within the usual percentages found concerning leads from other vendors. In the FOLLOWPACE study, the observed rate of lead failure was 4.0% during a mean follow-up of 5.8 years [5]. A more recent publication found an unexpected high rate of electrical abnormalities of 9.4% within three years concerning Beflex^®^ and Vega^®^ leads (MicroPort), in contrast to the 5076 CapSureFix Novus^®^ lead (Medtronic) in which electrical abnormalities were detected in only 2.2% during 3 years of follow-up [12].

Due to a rather systematic chronic loss of impedance in our cohort, we suppose that this phenomenon does not indicate lead fracture but might be a distinct feature of the Osypka QT-5^®^ ventricular lead. Importantly, our findings suggest that lead revision can be safely avoided or postponed in selected patients as long as R-wave sensing problems or insufficient pacing have not been encountered. However, this study has several limitations. The retrospective study design did not allow for a complete work-up of both pacemaker follow-up examinations and reasons for death in all patients. Thus, we did not focus on the clinical implications of the decline in lead impedance.

In conclusion, this study provides long-term data of patients implanted with the Osypka QT-5^®^ ventricular lead, precisely reporting on lead sensing, pacing amplitude and impedance during a median of 4 years of follow-up. To our knowledge, this is the first study to report on a gradual loss of impedance in this type of lead being not directly associated with severe clinical complications or a reduced survival rate. Although caution and stringent follow-ups are warranted in patients implanted with this lead showing decreased impedance, lead revision can be avoided in selected patients. Further studies are needed to fully investigate this phenomenon of the Osypka QT-5^®^ ventricular lead.

## Figures and Tables

**Figure 1 jcm-10-00639-f001:**
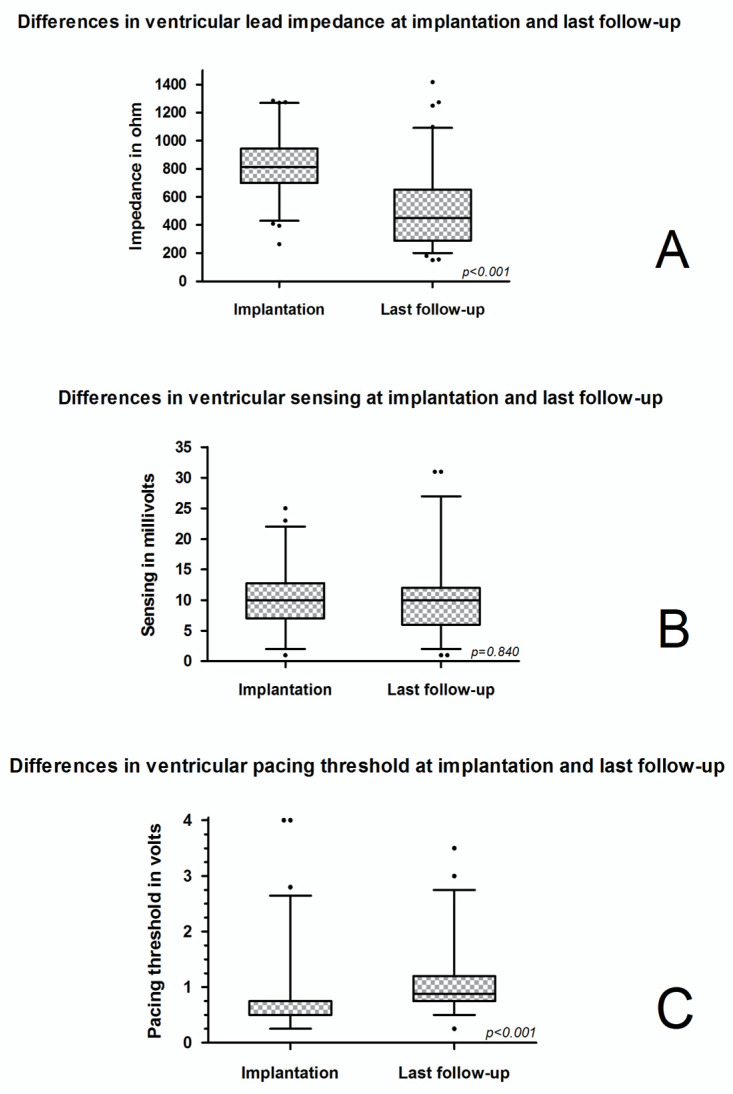
Device characteristics (lead impedance, sensing and pacing threshold) at implantation and at last follow-up: (**A**) differences in ventricular lead impedance at implantation and at last follow-up; (**B**) differences in ventricular sensing at implantation and at last follow-up; (**C**) differences in ventricular pacing threshold at implantation and at last follow-up. Whiskers in box plots mark the 95% (2.5–97.5%) confidence interval.

**Figure 2 jcm-10-00639-f002:**
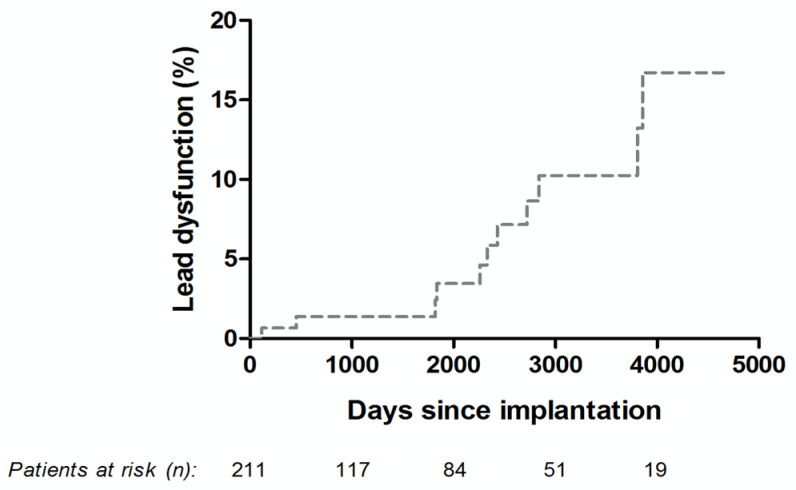
Kaplan-Meier curve describing the prevalence of dysfunction of the Osypka QT-5^®^ ventricular lead.

**Table 1 jcm-10-00639-t001:** Baseline characteristics.

Characteristics	
Age, years (IQR)	78.2 (72.0–83.5)
Male, *n*/%	109/51.7%
Height, cm (IQR)	168 (160–175)
Weight, kg (IQR)	76.0 (66.0–86.0)
Body surface area, cm^2^	1.9 (1.7–2.0)
Body mass index	27.0 (24.1–29.4)
Hypertension, *n*/%	162/76.8%
Diabetes, *n*/%	53/25.1%
Chronic kidney disease, *n*/%	42/19.9%
Coronary artery disease, *n*/%	83/39.3%
Previous stroke, *n*/%	19/9.0%
Left bundle branch block, *n*/%	16/7.6%
Right bundle branch block, *n*/%	38/18.0%
First pacemaker implantation, *n*/%	195/92.4%
Primary indication	
Higher degree atrioventricular block, *n*/%	79/37.4%
Sick sinus syndrome, *n*/%	47/22.2%
Higher degree atrioventricular block + sick sinus syndrome, *n*/%	7/3.3%
AF with bradycardia, *n*/%	55/26.1%
Bradycardia-tachycardia syndrome, *n*/%	23/10.9%

Abbreviations: cm, centimeter; IQR, interquartile range; kg, kilogram; *n*, number; %, percentage.

**Table 2 jcm-10-00639-t002:** Perioperative parameters.

Characteristics	
DDD-R *n*/%	139/65.9%
VVI-R *n*/%	72/34.1%
Pacemaker implanted ^1^	
Sensia^®^, *n*/%	43/20.4%
Adapta^®^, *n*/%	42/19.9%
Nexus^®^, *n*/%	37/17.5%
Zephyr^®^, *n*/%	17/8.1%
Altrua^®^, *n*/%	16/7.6%
Reply^®^, *n*/%	15/7.1%
Cylos^®^, *n*/%	10/4.7%
Other, *n*/%	31/14.7%
Right atrial lead ^2^	
4592 CapSure SP Novus^®^, *n*/%	123/88.5%
1944 IsoFlex^®^, *n*/%	6/4.3%
Other, *n*/%	10/7.2%
Primary pacing mode programmed	
DDD, *n*/%	138/65.4%
VVI, *n*/%	73/34.6%
Perioperative complications	
Dislocation, *n*/%	4/1.9%
Hematoma, *n*/%	9/4.3%
Pneumothorax, *n*/%	7/3.3%
Infection, *n*/%	2/0.9%
Loose set screw, *n*/%	1/0.5%

Abbreviations: DDD-R, dual-chamber rate-adaptive pacemaker; IQR, interquartile range; *n*, number; VVI-R—ventricle-paced, ventricle-sensed rate-adaptive pacemaker%, percentage. ^1^ Devices include: Sensia^®^ SESR01 (Medtronic, Dublin, Ireland), Adapta^®^ ADDR01 (Medtronic, Dublin, Ireland), Nexus^®^ I Ultra DR 1490 (Boston Scientific, Marlborough, MA, USA), Zephyr^®^ XL DR 5826 (St. Jude Medical, Little Canada, MN, USA), Altrua^®^ 60 (Boston Scientific, Marlborough, MA, USA), Reply DR^®^ (MicroPort, Shanghai, China; formerly Sorin/LivaNova), Cylos 990^®^ DR-T (Biotronik, Berlin, Germany). ^2^ Right atrial leads include: 4592 CapSure SP Novus^®^ (Medtronic, Dublin, Ireland), 1944 IsoFlex^®^ (St. Jude Medical, Little Canada, MN, USA).

**Table 3 jcm-10-00639-t003:** All reoperations during long-term follow-up.

Any reoperation, *n*/%	78/37.0%
Any reoperation without revision of the ventricular lead, *n*/%	42/19.9%
Battery replacement, *n*/%	36/17.1%
Upgrade to cardiac resynchronization therapy with pacemaker, *n*/%	4/1.9%
Dislocation of atrial lead, *n*/%	1/0.5%
Chronic pocket pain, *n*/%	1/0.5%
Any reoperation with revision of the ventricular lead, *n*/%	36/17.1%
Battery replacement with precautionary revising of the ventricular lead due to low impedance, *n*/%	5/2.4%
Upgrade to cardiac resynchronization therapy with defibrillator and necessity to implant an ICD lead, *n*/%	6/2.8%
Dislocation of ventricular lead, *n*/%	4/1.9%
Infection of pacemaker system, *n*/%	2/0.9%
Atrial lead dysfunction with revision of ventricular lead, *n*/%	2/0.9%
Low impedance of ventricular lead, *n*/%	2/0.9%
Dislocation of atrial and ventricular lead, *n*/%	2/0.9%
Cut of ventricular lead during tricuspid valve repair, *n*/%	1/0.5%
Loose set screw, *n*/%	1/0.5%
Ventricular lead dysfunction, *n*/%	11/5.2%

**Table 4 jcm-10-00639-t004:** Characteristics of patients with lead dysfunction.

Days after Implant	R-Wave Sensing (mV)	Lead Impedance (Ω)	Pacing Threshold (V/ms)	Malfunction and Consequence
3858	Not measurable	>3000	Not measurable	Lead fracture/lead replacement
2429	10.0	730	2.10 at 0.4 ms	Increasing pacing threshold/replaced as part of battery exchange
3807	Not measurable	200	0.75 at 0.4 ms	Impedance drop/replaced as part of battery exchange
1834	5.9	280	Not measurable	Exit block/lead replacement
2259	7.2	240	1.50 at 0.4 ms	Artefact oversensing and impedance drop/lead replacement
2840	1.5	200	Not measurable	Impedance drop and exit block/lead replacement
455	12.0	200	0.75 at 0.4 ms	Impedance drop/lead replacement
2720	5.4	350	Not measurable	Artefact oversensing and exit block/lead replacement
116	Not measurable	>3000	Not measurable	Lead fracture/lead replacement
2329	6.3	200	Missing value	Impedance drop/replaced as part of battery exchange
1821	3.7	259	2.10 at 0.4 ms	Impedance drop and increasing pacing threshold/replaced as part of battery exchange

## Data Availability

The data presented in this study are available on request from the corresponding author.

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
