# Peer review of "Long-Term Technical Performance of the Osypka QT-5® Ventricular Pacemaker Lead"

_jcm, 2021, doi:10.3390/jcm10040639_

Round 1
Reviewer 1 Report
Satisfactory
Wordy and could be shortened.
Tables are repeated in text.
Otherwise acceptable
Author Response
First of all, I want to express my sincere appreciation for your reviews. The ideas and corrections prevailed have significantly increased the value of the manuscript. Therefore I hope my answers to the reviewers are satisfying and fulfill the expectations for publication.
Reviewer 1
Reviewer: Satisfactory. Wordy and could be shortened. Tables are repeated in text. Otherwise acceptable.
Answer: Thank you for your valuable feedback. We shortened the section “3.1 patient characteristics” by removing the description of co-morbidities and pacing indications, which are now only described in Table 1.
Reviewer 2 Report
I would like to thank the authors for their reply and the way they much improved the manuscript i read with much interest.
- Now the way the re intervention on the lead are presented is much clearer in the table. However, there still a point to clarify :
In the « Any reoperation with revision of the ventricular lead, » heading, « Battery replacement with precautionary revising the ventricular lead due to low impedance » are clearly shown and occur in 5 patients. However, in the table 4 including ventricular lead dysfunction, some patients presented impedance drop and underwent lead replacement as part of battery exchange.
In these 2 cases, it seems patients had low impedance at the time of battery replacement and that lead was also replaced at the same time. Therefore, maybe the patients with battery replacement and low impedance in table 3 should be added to table 4. Was the clinical presentation different? can the author calrify this point?
- I understand that the authors state that chronic loss of impedance might be a distinct feature of the 247 Osypka QT-5® ventricular lead but the clinical relevance can be questionned.
Even if the study is retrospective, and to bring further insight into this point, i would recommand authors to split the cohort according the impedance loss during follow-up (e.g based on tertiles) and to describe the pacing threshold, R wave sensing in each group and the proportion of lead disruption.
If these parameters evolve in the same proportion irrespective of impedance loss, it would be in favor of the « wait and see » attitude. Otherwise, this would highlights patients pattern with particular risk for lead dysfunction.
Author Response
Please see the attachment.
Reviewer 2
Reviewer: In the « Any reoperation with revision of the ventricular lead, » heading, « Battery replacement with precautionary revising the ventricular lead due to low impedance » are clearly shown and occur in 5 patients. However, in the table 4 including ventricular lead dysfunction, some patients presented impedance drop and underwent lead replacement as part of battery exchange.
In these 2 cases, it seems patients had low impedance at the time of battery replacement and that lead was also replaced at the same time. Therefore, maybe the patients with battery replacement and low impedance in table 3 should be added to table 4. Was the clinical presentation different?
Can the author clarify this point?
Answer: Thank you for addressing this lack of ambiguity. Indeed, 5 patients underwent revision due to decreasing impedance, although lead parameters were still in normal range. This was due to the fact, that these patients required frequent ventricular pacing putting them at an increased risk in case of lead dysfunction. To clarify this issue, we adapted the wording from “precautionary revising the ventricular lead due to low impedance” to “precautionary revising the ventricular lead due to decreasing impedance and frequent ventricular pacing”.
Reviewer: I understand that the authors state that chronic loss of impedance might be a distinct feature of the 247 Osypka QT-5® ventricular lead but the clinical relevance can be questioned. Even if the study is retrospective, and to bring further insight into this point, i would recommand authors to split the cohort according the impedance loss during follow-up (e.g based on tertiles) and to describe the pacing threshold, R wave sensing in each group and the proportion of lead disruption.
If these parameters evolve in the same proportion irrespective of impedance loss, it would be in favor of the « wait and see » attitude. Otherwise, this would highlights patients pattern with particular risk for lead dysfunction.
Answer: Thank you for raising this issue. As recommended, we calculated three tertiles by using the decrease of impedance between first and last pacemaker check-up. There were no statistically significant differences found regarding R-wave sensing and ventricular pacing threshold. This analysis was implemented into the results section, methodology (statistics) was adapted to display the
according tests.

Reviewer 3 Report
This is a single-center, retrospective, observational study which summarizes the device performance of the Osypka QT-5® ventricular lead for pacemaker over a 4-year period. In 211 patients, there were 11 (5.2%) lead revisions due to mechanical lead dysfunction, although the median electrical impedance was halved at the last follow-up compared with the time of implantation. Thus, the authors conclude that the use of this lead is clinically practical.
The results of the paper are presented concisely, are comparable with other literature and the discussion is consistent with the data presented. I have no objection to the content of the paper. Nevertheless, there are some points that are unclear and need to be revised.
Line 82: Specify the source of the definition of lead dysfunction.
Line 106: Aren't there 42 patients with chronic kidney disease, 19.9% of the total?
Line 148: What is the difference between the four-year follow-up period for reoperation and the period described in this paragraph?
Table 1:
- Is the “+” following higher degree atrioventricular block a mistake for n/%?
- There's a duplicate row for "Sick sinus syndrome".
- The "N" in Abbreviations is in lower case (also Table 2).
Table 2: Please unify the ® notation.
Table 4: The text is written in American English, so it should be changed to "Artifact".
Author Response
Reviewer 2
Reviewer: This is a single-center, retrospective, observational study which summarizes the device performance of the Osypka QT-5® ventricular lead for pacemaker over a 4-year period. In 211 patients, there were 11 (5.2%) lead revisions due to mechanical lead dysfunction, although the median electrical impedance was halved at the last follow-up compared with the time of implantation. Thus, the authors conclude that the use of this lead is clinically practical.
The results of the paper are presented concisely, are comparable with other literature and the discussion is consistent with the data presented. I have no objection to the content of the paper. Nevertheless, there are some points that are unclear and need to be revised.
Answer: Thank you for review. Your feedback is very much appreciated.
Reviewer: Line 82: Specify the source of the definition of lead dysfunction.
Answer: Unfortunately, there is no direct source or generally a universal definition. As in previous studies, the definition was indicated by authors’ discretion.
Reviewer: Line 106: Aren't there 42 patients with chronic kidney disease, 19.9% of the total?
Answer: Thank you very much for pinpointing this error. Corrections were applied as indicated.
Reviewer: Line 148: What is the difference between the four-year follow-up period for reoperation and the period described in this paragraph?
Answer: Thank you for this comment. These two periods differ as one of them aims to report the time to any reoperation and the other one aims to report the time to occurrence of death. If patients required a reoperation their follow-up ended at that time point (censored due to event), but they were kept in analysis for survival.
Reviewer: Table 1:
- Is the “+” following higher degree atrioventricular block a mistake for n/%?
- There's a duplicate row for "Sick sinus syndrome".
- The "N" in Abbreviations is in lower case (also Table 2).
Table 2: Please unify the ® notation.
Table 4: The text is written in American English, so it should be changed to "Artifact".
Answer: Thank you for your corrections. The “+” symbol has been replaced by “and” as there were some patients suffering from both entities. The duplicate row has been deleted. The “N” abbreviation was lowered in both cases. The “®” notation was unified and the wording was changed to “Artifact” in table 4.
Round 2
Reviewer 2 Report
I would like to thank the authors for their reply.
I have no further comment regarding the current version of the paper.
This manuscript is a resubmission of an earlier submission. The following is a list of the peer review reports and author responses from that submission.
Round 1
Reviewer 1 Report
The manuscript “Long-term technical performance of the Osypka….” By Semmler et al reviews the performance of a ventricular pacemaker lead.
I have a number of comments:
Although the manufacturer is long established, most readers would not be familiar with the lead as the company is small.
The introduction and references relate to the indications for pacing. This is not relevant to the topic. What is the relevance of leadless pacemakers?
There are a number of abbreviations not defined; BMI, CIED.
All products mentioned need an address, but only once. Medtronic address is given three times.
This is an evaluation of a ventricular pacing lead. Documenting atrial leads and pulse generators should be in a table, only to prevent confusion with data.
Was this a prospective or retrospective study?
The time of day of the implant has been recorded. This is not relevant unless it is to be included in the discussion.
There is no discussion on the possible causes of the low impedance at follow-up. Does this occur with other manufacturers and references are required? The very low impedance suggests insulation breakdown. The insulation material and lead design are obviously essential in the discussion. The lead complications may all be related to the implanting techniques of the operators. Hence the importance of a control group if prospective. I assume that a loose set screw is operator failure to attach the pulse generator.
Over the study period, the lead was explanted in 36 patients, but lead dysfunction was documented in 11. Why were the other leads explanted? Were the leads removed or abandoned? Were the leads explanted because of the low impedance? Early large area cathodes in the 1960/70’s frequently had impedance values of 200 ohm measured at surgery or later by telemetry. Current drain was high, but function overall satisfactory. Did the operators wait until the power source was depleted? What are the recommendations for the readers, as this is the main finding in the manuscript? This is briefly discussed but is the most important part of the manuscript. Did the implanters discuss the results with the manufacturer?
Reviewer 2 Report
I have read with a great interest the manuscript written by Semmler et al.
This paper focus on a specific pacemaker lead (Osypka QT-5) and its related dysfunctions ocurring over the time and provides a long term follow-up of 211 patients.
The number of patients included reflects a high volume center with a high experience in pacemaker implantation. However i have some concerns regarding the manuscript :
Major concerns :
1. The number of lead dysfunction is not that clear. Indeed, authors report a total of 11 patients with lead dysfunction who required lead replacement but in the same paragraph, it is reported that : « During a median follow-up period of 4.0 years (IQR 0.1 - 8.0), 78 patients (37.0%) required any reoperation with the major indication being battery replacement due to depletion in 44 patients (20.9%). Other indications included any technical issues with the Osypka QT-5® ventricular lead (n = 12, 5.7%), the atrial lead (n = 3, 1.4%) or both leads (n = 4, 1.9%) »
So in this sentence, 19 patients seem to have technical issues with Osypka lead. Can the author explain that discrepancy ?
2. In the table 3 providing detail in patients with lead dysfunction, 3 patients underwent lead replacement due to impedance drop. Lead replacement for impedance drop is quite unusual and details regarding the way the impedance drop in such patient would be of interest (i.e to see if these lead replacement were done in the setting of abrupt vs progressive impedance decrease)
3. In the same table, can the author clarify if the reported pacing threshold correspond to pacing threshold at the time of lead dysfunction ? R-wave sensing is also reported as not measurable for some patients. Can the author clarify if this « not measurable R wave » is due to loss of sensing or a low ventricular escape rhythm ?
4. It is stated in the discussion that « the slight but significant increase in pacing threshold did not affect output programming and battery longevity and was detected only very late in the retrospective analysis of the database ». However the authors do not provide changes in pacemaker programming over the time to support that statement.
5. The major finding here is that this lead is associated with a progressive decline in impedance loss over the time. However, the clinical signifiance of that point is unclear. Indeed some lead were explanted because of impedance drop without any other dysfunction. This point should be further discuss.
Minor comments :
Result paragraph : i guess CIED is cardiovascular implantable electronic devices but it is not clearly stated.
Reviewer 3 Report
In the present manuscript, Semmler et al present a cohort of 211 patients who underwent an initial Osypka QT-5 ventricular lead implantation during a 7-year period (2006-2012). In this very detailed description, they report an overall drop in Impendance from 1142 Ohm, post implantation, to 420 Ohm after a median 5.2 years follow-up period. The manuscript is well-written and well-structured. However, there are major issues with regard to the study findings and their interpretation.
My comments to the authors are:
- Firstly, I would like to highlight that these types of leads are practically not in use any more. Therefore, the significance of the present study is restricted by this fact and its clinical relevance is probably low.
- Moreover, the manuscript, although well-written, it contains too many unnecessary details. In its present extended form, it could be published in a local journal for audit purposes. Alternative, it could be published in the form of ‘’a letter to the editor’’ or ‘’short communication’’ in a journal with a special interest for Pacing and Electrophysiology, however aiming to present only the most important findings, namely the drop in Impendance and possibly the assumed technical disadvantages of the lead.
- Despite the progress in pacing technology, pacemaker implantation may still be associated with significant peri- and/or post-procedural complications, with the transvenous lead remaining the weakest ring at the chain of pacing. Overall, the rate of peri-procedural and early post-procedural complications therapy is estimated at approximately, 10% (Kirkfeldt et al. Eur Heart J. 2014;35:1186-1194 & Udo et al. Heart Rhythm. 2012;95:728-735). In addition, chronic lead failure (due to insulation problems, lead fracture etc) was reported to be 15% (Kleeman et al. Circulation 2007;115:2474-2480). Therefore a presumable 17% chronic lead failure for Osypka QT-5 would not be a surprisingly increased rate.
- In line 186, authors state: ‘’Overall, the Osypka QT-5® ventricular lead was explanted in 36 patients (17.1%)’’. Also in line 195, authors mention that: ‘’This led to a significant rate of lead explantations (17.1%) during long-term follow-up’’. However, regarding the methodology followed in the study, it appears that the authors come to this conclusion due to the addition of various causes of explantation (infection, lead dysfunction, dislocation, other causes). However, the nature of causes such as lead dysfunction, explantation due to an infection or dislocation due to loose set screw is totally different. Moreover, as far as I can understand, there were 2 patients with lead infections, and 5 patients with lead dislocation as well as 11 patients with lead dysfunction (fracture of the lead or insulation problems, exit block etc). It is unclear what was the reason for lead explantation in the remaining 18 patients (ie since overall, the lead was explanted in 36 patients). In any case, the rate of 5.2% of lead dysfunction for this population is fairly acceptable.
- The study demonstrates a significant decrease in lead impendance overtime. However, I should observe that the baseline mean impendance value (1142 Ohm- IQR: 955-1285 Ohm) is probably higher than we would expect. In addition, a follow-up value of 450 Ohm (IQR: 289-652) is within the acceptable limits, in contrast to the Impendance values of 200 Ohm in Table 3 that are probably due to insulation problems.
- Authors conclude that: ‘’…our findings suggest that lead explantation can be safely avoided or postponed in selected patients as long as R-wave sensing problems or insufficient pacing have not been encountered’’. Do they suggest a lower cut off value in these cases? I don’t think that we can generalize these findings, since this may be a inhomogeneous population including patients with lead dysfunction as well as patients with impendance value within normal limits. Is it wise to suggest a ‘’wait and see’’ approach, for all cases, irrespectively to impendance values?
